

# Heart failure survival prediction using novel transfer learning based probabilistic features

Azam Mehmood Qadri[1], Muhammad Shadab Alam Hashmi[1], Ali Raza[1], Syed Ali Jafar Zaidi[2] and Atiq ur Rehman[3]

[1] Institute of Computer Science, Khwaja Fareed University of Engineering and Information Technology, Rahim Yar Khan, Pakistan

[2] Institute of Information Technology, Khwaja Fareed University of Engineering and Information Technology, Rahim Yar Khan, Pakistan

[3] Artificial Intelligence and Intelligent Systems Research Group, School of Innovation, Design and Engineering, Mälardalen University, Västerås, Sweden

## ABSTRACT

Heart failure is a complex cardiovascular condition characterized by the heart's inability to pump blood effectively, leading to a cascade of physiological changes. Predicting survival in heart failure patients is crucial for optimizing patient care and resource allocation. This research aims to develop a robust survival prediction model for heart failure patients using advanced machine learning techniques. We analyzed data from 299 hospitalized heart failure patients, addressing the issue of imbalanced data with the Synthetic Minority Oversampling (SMOTE) method. Additionally, we proposed a novel transfer learning-based feature engineering approach that generates a new probabilistic feature set from patient data using ensemble trees. Nine fine-tuned machine learning models are built and compared to evaluate performance in patient survival prediction. Our novel transfer learning mechanism applied to the random forest model outperformed other models and state-of-the-art studies, achieving a remarkable accuracy of 0.975. All models underwent evaluation using 10-fold cross-validation and tuning through hyperparameter optimization. The findings of this study have the potential to advance the field of cardiovascular medicine by providing more accurate and personalized prognostic assessments for individuals with heart failure.

# INTRODUCTION

The primary cause of heart failure (HF) is coronary artery disease (CAD), often precipitated by arterial blockages leading to heart attacks. Heart disease or high blood pressure is also associated with HF (*Gjoreski et al., 2017*). Various factors contribute to heart diseases, including recognized parameters such as alcohol intake, smoking, diabetes, high cholesterol, and a lack of exercise routine. Previous research has identified high blood sugar, poor diet, excess weight (*Benjamin et al., 2019*), and unhealthy activities as significant causes of heart disease. Elevated blood pressure thickens artery walls, impeding blood flow and contributing to a higher mortality rate (*Sugathan, Soman & Sankaranarayanan, 2008*).

Corresponding authors
Ali Raza, ali.raza.scholarly@gmail.com
Atiq ur Rehman,
atiq.ur.rehman@mdu.se,
atiqjadoon@gmail.com

Angina attacks are another indicator of heart disease (*Bashir et al., 2019*). Diagnosing cardiac illnesses is a complex task requiring a multitude of information, tools, and laboratory tests (*Chen et al., 2018*). Many individuals undergo costly and resource-intensive tests that involve physical activity (*Gavhane et al., 2018*). Early and timely screenings are not always feasible, and accessing surgeries for heart diseases can be challenging. This is particularly true in economically developing countries where limited qualified health staff, diagnostic equipment, and facilities hinder proper treatment and accurate heart health diagnosis. Improving these conditions can help prevent heart attacks and enhance the overall well-being of patients.

Recent mortality states indicate that approximately twenty-six million people are affected by heart disease (*Savarese & Lund, 2017*). Without compulsory intervention, it is likely to reach its peak in the coming decades (*Benjamin et al., 2019*). Another recent study reported that around sixty-five million people have succumbed to cardiac diseases (*Virani et al., 2020*). Maintaining balanced nutrition and ensuring timely diagnosis are crucial for ensuring patient safety. Although angiography is considered the best and most accurate method for predicting heart disease, its high cost renders it inaccessible for many individuals (*Arabasadi et al., 2017*). Ample data and patient records from previous studies, as well as open-source access to hospital patient records, are available, according to experts who have assessed them (*Das, Turkoglu & Sengur, 2009*). Electronic health record systems prove beneficial for medical and investigative purposes in this modern era (*Chapman et al., 2019*). The potential for future loss of life due to minor medical examination errors related to heart disease underscores the importance of accurate diagnosis and treatment.

Research focusing on stages 3 and 4 of heart failure holds profound clinical significance as it addresses critical aspects of patient care and management. In these advanced stages, patients experience severe symptoms and a heightened risk of complications, necessitating targeted interventions. Medications such as beta-blockers, angiotensin-converting enzyme (ACE) inhibitors, and diuretics play pivotal roles in alleviating symptoms, optimizing cardiac function, and improving patients' quality of life. Beyond conventional considerations related to chronic kidney (CK) disease and diabetes, a comprehensive approach takes into account a broader array of relevant factors. These may include comorbidities, lifestyle factors, and socio-economic determinants that impact the overall health and well-being of heart failure patients.

Numerous computer technologies have been employed to mitigate the spread of diseases. It is widely recognized that machine learning (ML) has a profound impact on the medical field, offering various approaches for analyzing illnesses and predicting outcomes. Diagnosing cardiac illnesses and survival in patients is a complex task, requiring a myriad of information, tools, and laboratory tests rather than relying on traditional diagnostic methods (*Chen et al., 2018*). This study seeks to enhance prevalent technologies, specifically ML techniques, to detect heart failure (HF) survival and assess its likelihood. Utilizing machine learning, clinical experts can accurately identify survival rates related to heart disease, potentially reducing mortality rates (*Ansarullah & Kumar, 2019*). The integration of advanced algorithms holds significant promise in enhancing the precision and reliability of survival predictions, thereby contributing to personalized and effective

healthcare strategies. The research takes innovative strides in tailoring treatments for heart failure patients based on their medical histories, aiming to make substantial contributions to saving lives in the realm of heart failure management.

This study's key contributions to predict survival in heart failure patients are:

- A novel transfer learning-based feature engineering approach is proposed, which generates a new probabilistic feature set from patient data using the ensemble trees method, random forest.
- We have constructed nine fine-tuned machine learning models, including logistic regression, random forest, support vector machine, decision tree, XGBoost classifier, Gaussian naive Bayes, k-nearest neighbors, extra tree classifier, and gradient boosting classifier.
- The performance of the applied models has been validated through a 10-fold cross-validation process and tuning carried out *via* hyperparameter optimization.

This research study is further divided into different sections: 'Related Work' examines the literature on heart failure and provides overviews of previously conducted studies. 'Proposed Methodology' explains the applied methodology for heart failure, discussing the research design, data sources, data collection methods, and data analysis techniques. 'Results and Discussion' elaborates on the research outcomes of employed machine learning models and scientifically discusses our research approach with experimental results. 'Conclusion' of the research paper offers a concise summary of our research, summarizing the main key points and presenting the overall significance of our work.

## RELATED WORK

This section of our research explores the existing literature and studies in the field, providing a comprehensive overview of the advancements and methodologies applied to predict heart failure survival. Numerous studies have delved into heart failure prediction using traditional machine-learning approaches, as analyzed in Table 1.

*Mansur Huang, Ibrahim & Mat Diah (2021)* proposed to predict heart failure in patients by utilizing the UCI heart disease dataset. Employing a machine learning technique, the author developed a random forest model with 13 features. The achieved accuracy in heart failure prediction was 88 percent. Similarly, *Mamun et al. (2022)* focused on predicting survival in heart failure patients. The UCI HF dataset, comprising 299 patient records, was used for this analysis. Employing a machine learning approach, the author tested several models, with LightGBM identified as the optimal classifier. LightGBM demonstrated a superior accuracy score of 85 percent in predicting patient survival, outperforming other classifiers in the study.

*Newaz, Ahmed & Haq (2021)* proposed a technique aimed at preventing heart failure in patients. The dataset employed in this research originated from the HF clinical record dataset collected from the Allied Hospital of Cardiology in Faisalabad, comprising 299 patient records. The study utilized a machine learning approach and introduced an ensemble framework strategy to enhance the robustness of the random forest (RF) model, addressing data imbalance issues. Feature engineering involved the use of Chi-squared

**Table 1 Previous studies literature discussed.**

| Reference | Year | Dataset | Proposed technique | Feature engineering | Performance accuracy |
|---|---|---|---|---|---|
| *Mansur Huang, Ibrahim & Mat Diah (2021)* | 2021 | UCI-Heart disease | RF, ML | All 13 features | Accuracy 88% |
| *Mamun et al. (2022)* | 2022 | UCI-HF dataset | LIGHT-GBM, ML | correlation matrix | Accuracy 85% |
| *Newaz, Ahmed & Haq (2021)* | 2021 | HF-clinical record | RF classifier, ML | Chi2+BRF | Accuracy 76.25% |
| *Plati et al. (2021)* | 2021 | Multiple dataset | ROT, ML | All features | Accuracy 91.23% |
| *Hussain et al. (2020)* | 2022 | RR-interval time series | SVM-Linear kernel, ML | multimodal features | Accuracy 93.1% |
| *Guidi et al. (2014)* | 2020 | UCI-heart disease | Ensembled Voting based model (VBM) | 14 features | Accuracy 85.71% |
| *Javid, Alsaedi & Ghazali (2020)* | 2020 | UCI- repository | Hoeffding classifier, ML | 13 attributes | Accuracy 88.56% |
| *Kumar & Sikamani (2020)* | 2021 | Stanford online repository | KERAS, ML | attributes | Accuracy 80% |
| *Ashraf et al. (2021)* | 2021 | Cleveland heart dataset | RF, ML | 14- features | Accuracy 86.9% |
| *Mohan, Thirumalai & Srivastava (2019)* | 2019 | Cleveland heart dataset | HRFLM, ML | 13 attributes | Accuracy 88.7% |
| *Al-Absi et al. (2021)* | 2021 | Qatar Biobank dataset | CatBoost, ML | 150 features | 93% |
| *Ishaq et al. (2021)* | 2021 | Heart failure clinical record dataset | ETC, ML | RFE | Accuracy 92.62% |

and recursive feature analysis. The proposed random forest model demonstrated superior performance compared to other models, achieving an accuracy rate of 76.83 percent for predicting the survival of heart patients.

Similarly, *Plati et al. (2021)* established a systematic process for employing machine learning approaches to diagnose the presence of heart failure. This research is noteworthy for its impact on clinical procedures and its exploration of how different features influence classification correctness scores. Notably, when the entire set of attributes was employed for classification, the results for heart failure diagnosis exhibited excellent accuracy at 91.23 percent, sensitivity at 93.83 percent, and specificity at 89.62 percent. The findings of both studies contribute valuable insights to the field of heart failure prediction, with *Newaz, Ahmed & Haq (2021)* addressing data imbalance challenges through ensemble strategies and *Plati et al. (2021)* emphasizing the significance of feature selection in improving diagnostic accuracy.

*Hussain et al. (2020)* developed a method to automatically extract multi-model properties from heart rate variability (HRV) data, capturing its temporal, spectral, and dynamic characteristics. Robust machine learning algorithms, including support vector machine (SVM) with its kernel, decision trees (DT), k-nearest neighbors (KNN), and ensemble classifiers, were employed to evaluate detection performance. Performance metrics such as specificity and sensitivity were utilized to assess the algorithms. The SVM linear kernel exhibited excellent performance, achieving a correctness score of 93.1 percent, a sensitivity of 96 percent, and a specificity of 89 percent. This underscores the effectiveness

of the proposed method in accurately analyzing HRV data for comprehensive cardiac assessment.

*Javid, Alsaedi & Ghazali (2020)*, aiming to predict heart disease through the application of machine learning (ML) and deep learning (DL) approaches for enhanced accuracy. The research employed the UCI heart disease dataset to evaluate the effectiveness of ML and DL methods. A voting-based method was utilized to enhance the accuracy of weak classifiers, combining several algorithms. The proposed ensemble technique, employing a voting approach, achieved an accuracy of 85.71 percent with all attributes considered. Notably, this approach demonstrated a notable improvement of 2.1 percent in accuracy. Similarly, the author discussed another relevant study, *Kumar & Sikamani (2020)*, focused on predicting chronic heart disease. The UCI repository served as the dataset for this study, wherein a machine-learning approach was employed. Multiple classifiers, utilizing 13 attributes as indicators of the disease, were explored. The Hoeffding Classifier emerged as the top-performing classifier, achieving an impressive accuracy score of 88.56 percent in predicting chronic heart disease.

*Ashraf et al. (2021)* utilized deep learning technology to predict cardiovascular (CVD) disease. The dataset employed in this study was sourced from the Stanford online repository. Various forecasting approaches were applied to this dataset, including ensemble and learning methods. Among classifiers, J48 achieved a 70 percent accuracy score. The same dataset underwent analysis using a novel approach, incorporating TensorFlow, Keras, and PyTorch techniques. The results of the analysis indicated that J48 outperformed other models, boasting an 80 percent accuracy score in predicting CVD disease. Following a comprehensive analysis, the study concluded that both conventional and cutting-edge technologies present a novel approach for predicting CVD illness.

*Pal & Parija (2021)* suggested heart disease classification using the Kaggle Cleveland heart disease dataset, which consists of 14 features. Employing a machine learning approach, the study utilized a random forest model, achieving an accuracy of 86.9 percent. The evaluation concluded that the random forest classifier proves to be the most efficient for predicting heart disease. Similarly, *Mohan, Thirumalai & Srivastava (2019)* proposed an innovative approach to predicting heart disease. The dataset employed in this study was the Cleveland Heart dataset, comprising 13 features. A hybrid machine-learning approach was applied to optimize features and enhance the accuracy of heart disease prediction. The study utilized a novel hybrid machine learning approach to select optimal features, and a random forest linear model was employed for heart disease prediction, achieving an accuracy of 88.7 percent. The findings suggest that the hybrid model stands out as the most effective method for improving the accuracy of predicting heart disease.

*Al-Absi et al. (2021)* developed ML classifiers to distinguish cardiovascular disease from the control group using data obtained from the Qatar Biobank dataset (QBB). This dataset comprises 150 features extracted from various clinical records of residents in Qatar. Employing a machine learning approach, the study demonstrated that the proposed CatBoost algorithm outperformed other methods, achieving a remarkable accuracy of 93 percent, particularly excelling in the context of cardiovascular disease (CVD) detection.

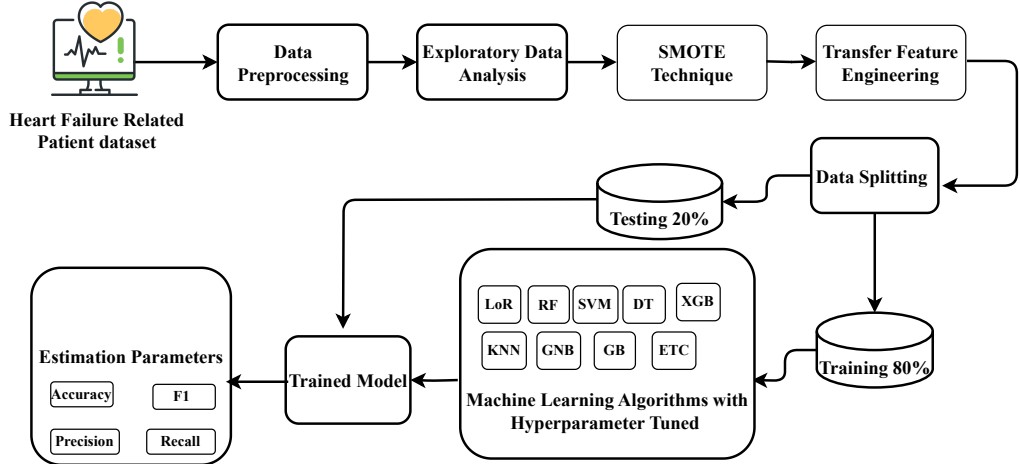

**Figure 1** Methodology workflow diagram analysis for targeted research.

In *Ishaq et al. (2021)*, the authors presented an effective approach for predicting heart failure patient survival. The dataset utilized in this research was the HF clinical record dataset downloaded from Kaggle, encompassing records from 299 patients. Employing a data mining approach, the study aimed to select optimal features to enhance the accuracy of predicting patients' survival. To address class imbalance, the Synthetic Minority Over-sampling Technique (SMOTE) was employed. Nine ML models were utilized to predict heart failure patient survival, with the Extra Trees Classifier (ETC) outperforming other models. It achieved an accuracy of 92.62 percent, using the highest-ranked features selected by RF to predict the survival of heart patients.

## PROPOSED METHODOLOGY

For heart failure survival prediction, this research utilized the HF clinical record dataset, which comprises records from two hundred and ninety-nine patients. The dataset underwent processing to ensure proper formatting. Exploratory data analysis was conducted to understand the data's structure and characteristics relevant to heart failure. The identification of data imbalance in the dataset presented a prediction challenge, which was addressed by applying the SMOTE to balance the dataset. Subsequently, a novel proposed transfer learning feature engineering method was employed on the balanced data, creating a new probabilistic feature set. Following this, the data was partitioned into training and testing phases, allocating 80 percent for training and 20 percent for testing the model on unseen data. Nine state-of-the-art machine learning approaches were employed, and constructed using the training data, and their performance was assessed on the unseen test data. The best-fit parameters were determined for the machine-learning approaches through hyperparameter tuning. The well-performing suggested classifier aims to forecast heart failure survival prediction with improved efficiency. The research methodology workflow diagram is illustrated in Fig. 1.

**Table 2  Heart failure dataset features discussion.**

| Sr # | Features | Description | Range | Measurement |
|------|----------|-------------|-------|-------------|
| 1 | Age | Patients age | 40–95 | Years |
| 2 | Anemia | Decrease of red blood cells or hemoglobin | 0,1 | Boolean |
| 3 | High blood pressure | If patient suffer in blood pressure | 0,1 | Boolean |
| 4 | Creatinine phosphokinase | Level of Creatinine phosphokinase in the blood | 23–7,861 | Mcg/L |
| 5 | Diabetes | Patient suffer in diabetes | 0,1 | Boolean |
| 6 | Ejection fraction | Percentage of blood leaving the heart at each contraction | 14–80 | Percnetage% |
| 7 | Sex | Men and women | 0,1 | Binary |
| 8 | Platelets | Platelets in blood | 25.01–850 | kiloplatelets/mL |
| 9 | Serum creatinine | Level of creatinine in blood | 0.50–9.40 | Mg/dL |
| 10 | Serum sodium | Sodium level in the blood | 114–148 | mEq/L |
| 11 | Smoking | If patient is smoker | 0,1 | Boolean |
| 12 | Time | Time period of follow-up | 4–285 | Days |
| 13 | Target Death Event | If the patient died during the follow-up period | 0,1 | Boolean |

## Dataset details

The HF Clinical Records Database (*Ahmad et al., 2017*) dataset is also available in the UCI Machine Learning Repository (*UCI Machine Learning Repository, 2020*). The dataset comprises health records of 299 patients with cardiac issues, and each individual profile includes thirteen clinical variables. There are 194 men, representing 64.88 percent, and 105 women, representing 35.12 percent, in the dataset. All patients are aged 40 or older. A label of 1 denotes a death event, while 0 denotes life. The dataset contains all values with no missing entries. The New York Heart Association (NYHA) classifies HF phases as III and IV. All patients had left ventricular systolic dysfunction and had previously experienced HF. The dataset details are presented in Table 2.

## Exploratory data analysis

To gain a deeper insight into the causes of HF, this section examines cardiac statistics and diverse dataset patterns. The proposed HF approach is employed to determine the significance level for the study, which concentrates on 13 variables and is used to train the model-based ML approach. These attributes are evaluated from different perspectives, and Matplotlib and count charts are employed for visualization.

Figure 2 charts display the overall quantity of examples in each group within the HF dataset. Figure 2 illustrates the gender (sex) distribution in the dataset, representing 0 for females (105, constituting 35.12 percent of the dataset) and one for males (194, constituting 64.88 percent).

The input attributes for the data are all numeric. Figure 3 presents the correlation evaluation of the heart failure dataset's attributes. According to the analysis, all features exhibit a robust relationship. Some attributes show low negative correlations, such as ejection fraction and serum sodium. Notably, only the feature "time" displays a strong negative correlation in the dataset. The analysis highlights a strong relationship among the

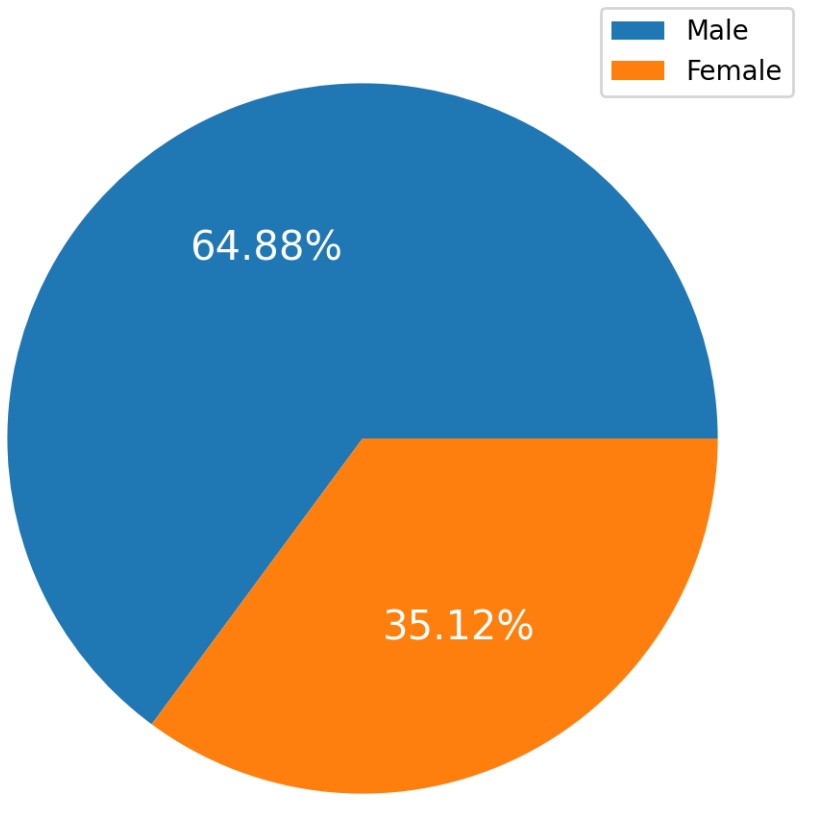

**Figure 2** **The overall quantity analysis.**

remaining dataset features. Some of the dataset attributes show low negative correlations, such as ejection fraction, serum sodium, and diabetes.

## Synthetic minority oversampling

SMOTE is a type of oversampling often utilized to address irregularities in data. The SMOTE technique, an example of an oversampling approach, has found widespread application in medical contexts to handle unbalanced class data (_Blagus & Lusa, 2015_). The SMOTE technique expands the number of samples of raw data by generating minority-class synthetic data randomly from its nearest neighbors. As these new samples are created based on actual information, they possess comparable attributes 53. It is important to note that SMOTE may introduce noise when applied to high-dimensional data, and its usage is discouraged in such cases. The SMOTE method is employed to construct a new training-balanced dataset. As illustrated in Fig. 4 and Fig. 5, SMOTE augments the quantity of samples for both imbalanced and balanced classes.

## Data splitting

We divided the data into training and testing phases. To apply the machine learning classifiers and generate forecast results on unseen data, we allocated 20 percent for testing

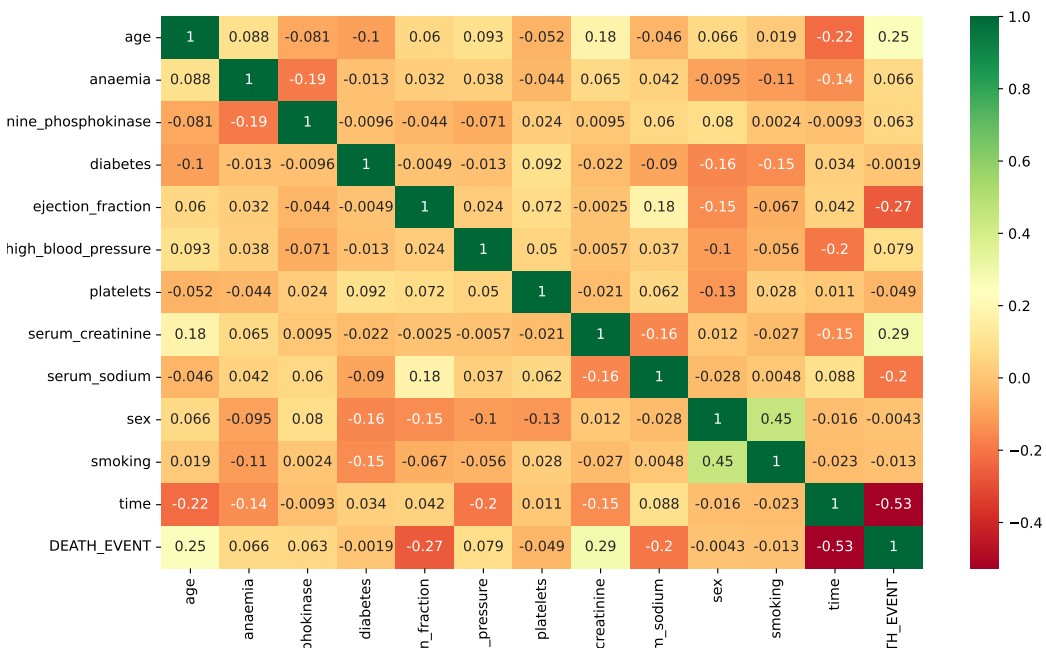

**Figure 3** The correlation evaluation of the heart failure dataset's attributes.

and 80 percent for training purposes. The machine learning classifiers utilized in this study are well-established and commonly employed in various learning problems.

## Applied machine learning classifiers

This section explores various machine learning approaches employed in predicting heart failure. It provides an explanation of how machine learning models function and introduces key terminology (*Zaidi, Tariq & Belhaouari, 2021*). Our proposed study evaluates nine advanced machine learning models for forecasting heart failure. Using supervised machine learning methods, the outcome of heart failure data is predicted.

### *Logistic regression*

Logistic regression (LoR) is a supervised statistical learning technique for classification and regression (*Daghistani & Alshammari, 2020*). LoR utilizes independent variables to forecast the categorical dependent variable. The binary classification probability measurements form the foundation for learning and prediction processes. In logistic regression models, class variables must be binary. Similar to the "Target" column in the dataset, this column consists of two binary numbers: 0 indicates patients unlikely to develop HF, and 1 denotes patients likely to develop HF.

### *Random forest*

The RF is a supervised machine-learning algorithm consisting of several decision trees (*Palimkar, Shaw & Ghosh, 2022*). The decision nodes of the tree represent the features, while the leaf nodes indicate the intended outcome. The final forecast is determined by a majority vote after using observations randomly selected by RF to construct decision

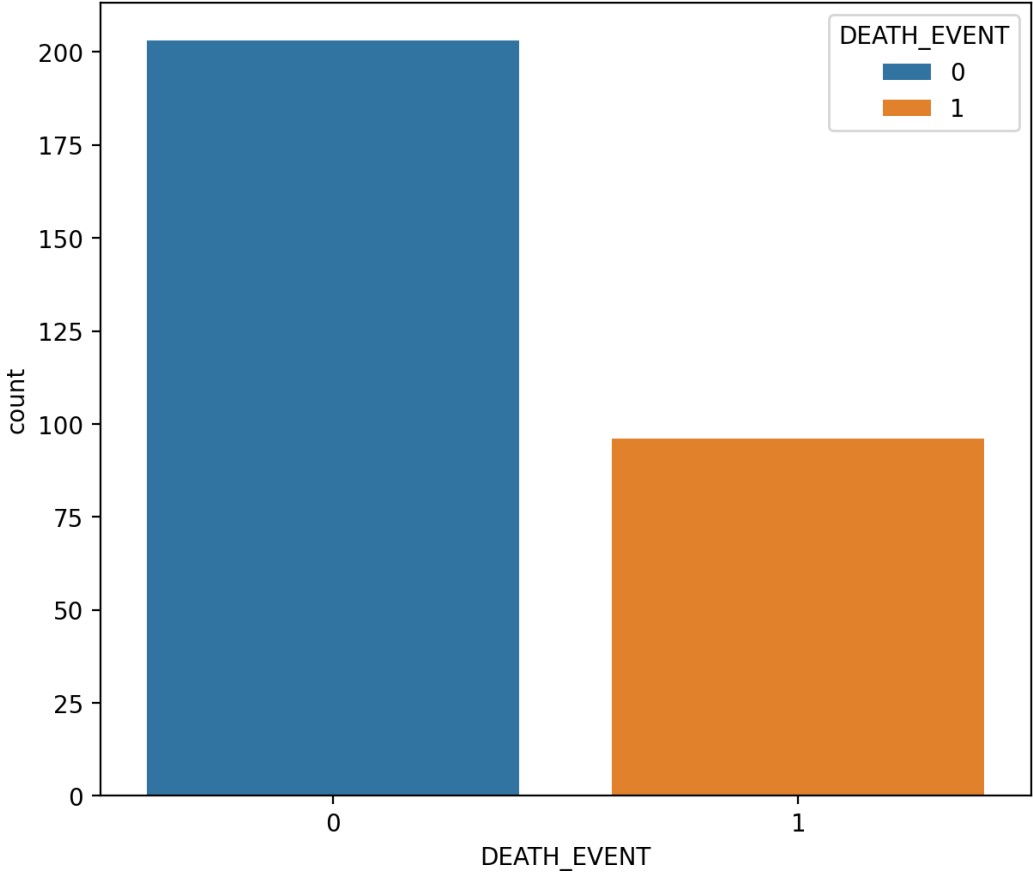

**Figure 4** The data analysis before balancing by SMOTE.

trees. RF is an ensemble learning technique that surpasses individual classifiers in terms of results. It enhances classification efficiency and mitigates overfitting issues.

### Support vector machine

SVM is a well-known supervised learning technique that may be applied to classification and regression issues (*Baldomero-Naranjo, Martinez-Merino & Rodriguez-Chia, 2021*). It performs best in classification problems. The goal of SVM is to offer suitable decision thresholds. In order to partition the data into target classes from the n-dimensional feature space, the SVM model creates a best-fit decision boundary (*ur Rehman, Khanum & Shaukat, 2013*). The decision boundary is known as a hyperplane. SVM chooses extreme vectors, or support vectors, to create the hyperplane. Because of this, the technique is often referred to as a support vector machine.

### K-nearest neighbor

KNN is a supervised learning method that predicts the data class by considering information from its closest neighbors. KNN attempts to group data points that are close to each other based on similarity (*Tajik et al., 2016*). This approach is non-parametric, categorizing data points according to their proximity. The training procedure is time-consuming due to the

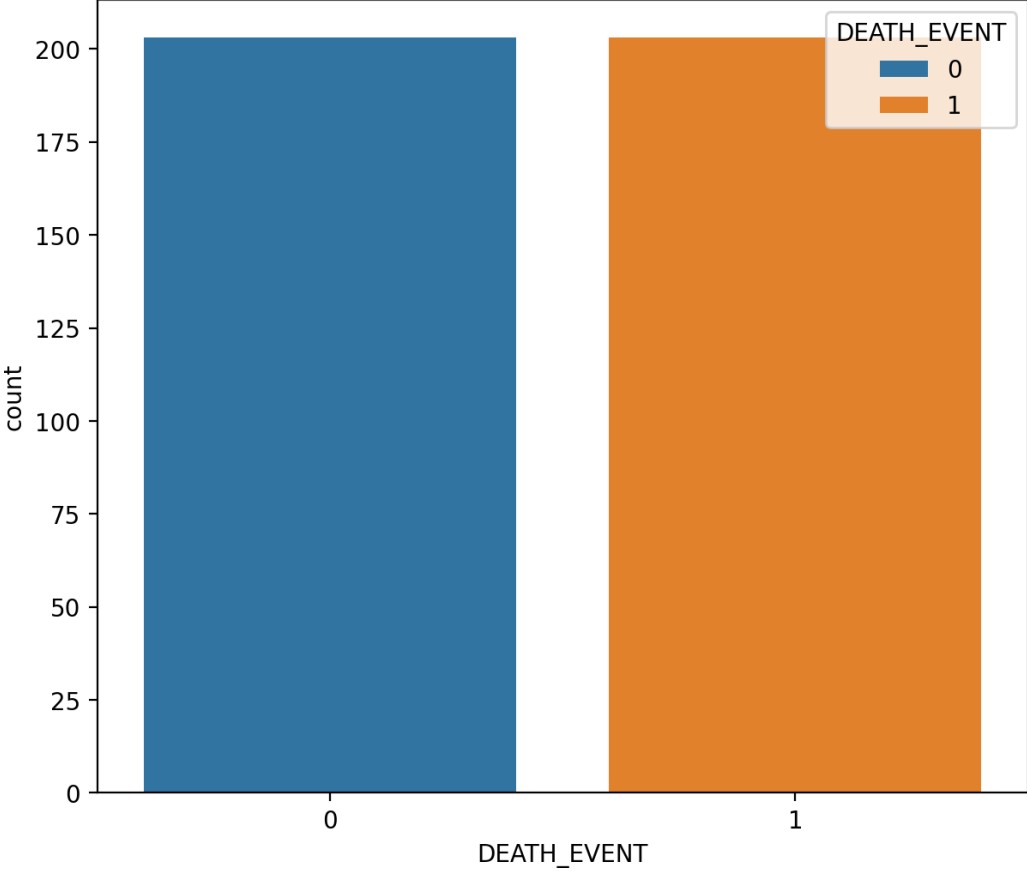

**Figure 5** The data analysis after balancing by SMOTE.

slow learning process. The similarity between data points is evaluated using metrics such as Euclidean distance or equivalent distance measures (*Jones & Hardiyanti, 2021*).

### Decision tree

An algorithm used for machine learning in classification problems is referred to as a DT (*Charbuty & Abdulazeez, 2021*). The tree-like structure of a DT comprises nodes and leaves, with data attributes allocated to inner nodes, and outcome labels stored in leaf nodes. In DT, the topmost node is the root node. Decision tree algorithms autonomously generate trees from input data, aiming to minimize generalization errors through techniques such as decision tree classification (DTC). The primary objective of DTC is to identify the optimal decision tree. A noteworthy challenge in decision tree construction is the selection of appropriate data properties.

### Extreme gradient boosting

A machine learning approach for supervised ensemble classification analysis is called Extreme Gradient Boosting (XG Boost) (*Fitriyani et al., 2020*). Ensemble learning algorithms combine various machine learning techniques to enhance performance. XGB is known for its adaptability, adequacy, and portability. It employs the parallel

gradient boosting tree method to address classification issues. To mitigate overfitting, XGB incorporates a superior regularization technique.

### Extra tree classifier

The ETC represents an advancement in the bagged decision tree-based ensemble learning approach (*Ossai & Wickramasinghe, 2022*). While the ETC and random forest share some underlying concepts, their distinction lies in the way the structure is generated. In the context of a classification task, the ETC amalgamates the outputs of diverse, unrelated decision trees to predict the target class. The ETC technique leverages the training data to generate multiple bagged decision tree samples, with the decision rule being chosen randomly. Predictions are then made using a majority voting approach based on the decision trees. The outcomes of the majority voting process are aggregated to produce the final forecast.

### Gradient boosting classifier

The gradient boosting (GB) technique is the most widely employed progressive-learning ensemble technique (*Rufo et al., 2021*). Predictive analytics proves effective when utilizing both regression and classification. The GB approach progresses incrementally (*Bowd et al., 2020*). By amalgamating the outcomes of numerous weak models, we can construct a final predictive model that accurately forecasts. The GB technique aims to amalgamate multiple weak models into a robust one. GB constructs a model sequentially, training each primary classifier individually. The goal is to establish a reliable model. A weak model can transform into a valuable asset through the integration of numerous models.

### Gausian Naïve Bayes

The Gaussian naive Bayes (GNB) supervised machine learning algorithm was designed (*Barus et al., 2020*). The GNB model is based on the naive Bayes theorem and associated methodologies. The GNB approach (*Cataldi, Tiberi & Costa, 2021*) assumes that all predictors are independent, which is a strong premise. It posits that one feature of a class can exist separately from another part of the class. The GNB utilizes a Gaussian distribution and naive assumptions to forecast the target class.

## Novel proposed transfer learning

A novel feature engineering technique is proposed in our research, as shown in Fig. 6. Our projected approach extracts class probability features (*Raza et al., 2023*) by inputting the heart failure clinical record dataset. Our research experiments revealed that the suggested feature engineering performs best in heart failure survival prediction scores. The random forest model is trained using a training set, and the random forest technique, employing the function `predict_probability()`, predicts the class probability. Here, $X$ denotes the input data attributes, and $y$ represents the target labels.

$$\mathcal{D}_{\text{train}} = \{(X_1, y_1), (X_2, y_2), \ldots, (X_n, y_n)\} \tag{1}$$

where the total number of training samples is represented by $n$.

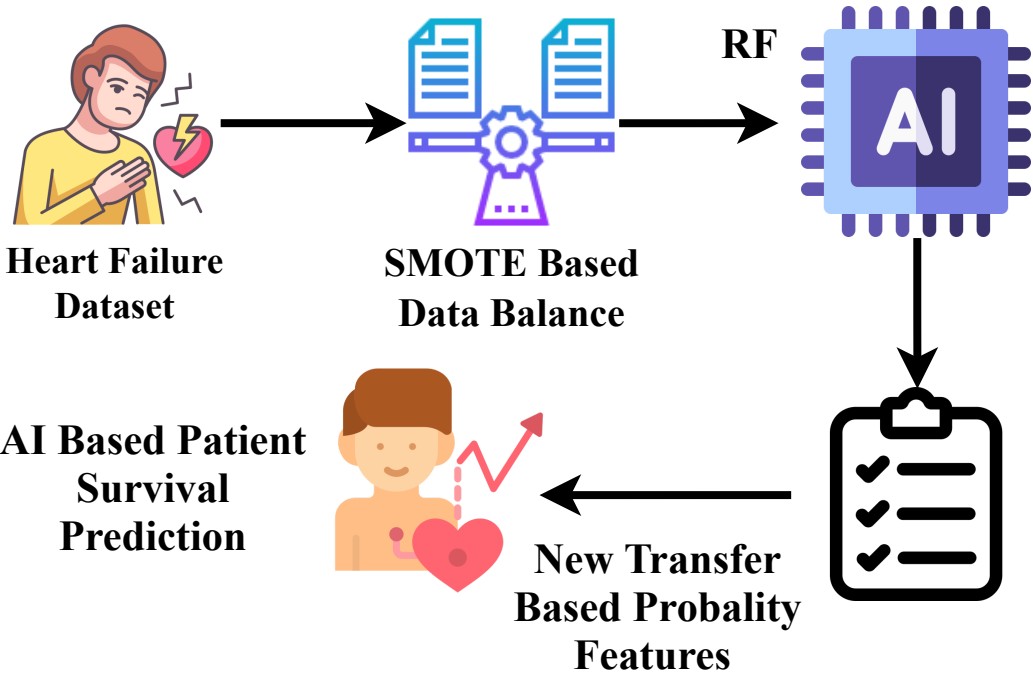

**Figure 6** **Workflow diagram analysis of novel proposed transfer learning apporach.** Disease icons created by B.Genesis - Flaticon. Scale icon created by Eucalyp - Flaticon. AI icon by Freepik. Survival icon created by Good Ware - Flaticon. Test icons created by Freepik - Flaticon.

The random forest model consists of a collection of decision trees, each constructed using a random subset of the training data. When given a novel input sample, $X_{new}$, the random forest predicts the probabilities of its class by aggregating the predictions from each decision tree. $T_1, T_2, \ldots, T_i$ represent the distinct decision trees within the random forest, with $i$ denoting the total number of trees. For a specific tree $T_i$, the computation of class probabilities for $X_{new}$ is as follows:

$$P_{\text{tree},i}(X_{new}) = \text{class probabilities for } X_{new} \text{ in } T_i \tag{2}$$

The aggregated predicted class probabilities of the random forest model result from averaging the predictions made by each tree.

$$P_{\text{ensemble}}(X_{new}) = \frac{1}{m} \sum_{i=1}^{m} P_{\text{tree},i}(X_{new}) \tag{3}$$

In this context, $N_{\text{class},i}(X_{new})$ indicates the number of data points in the leaf node of $T_i$ associated with a particular class, and $N_{\text{samples},i}$ represents the total sample count in that leaf node. The function `predict_proba()` in random forest libraries computes and provides $P_{\text{ensemble}}(X_{new})$, presenting the class probabilities for the input sample $X_{new}$.

## Hyperparameter tuning

Appropriate training and testing processes (*Isabona, Imoize & Kim, 2022*) are employed to determine the optimal hyperparameter values for applied machine learning models.

**Table 3  Evaluation of optimal hyperparameter for our proposed study using machine learning classifiers.**

| Models | Hyperparameters |
| --- | --- |
| LoR | random_state=1 |
| RF | max_depth=300,n_estimators=300, random_state=0 |
| SVM | Kernal= Sigmoid |
| DT | random_state=2, criterion='gini', max_depth=100, min_samples_split=2 |
| XGB | use_label_encoder=False, eval_metric= mlogloss |
| GNB | priors=None, var_smoothing=1e−09 |
| KNN | n_neighbors=9 |
| ETC | n_estimators=300, random_state=2, max_depth=300, min_samples_split=2 |
| GB | n_estimators=100, learning_rate=1.0, max_depth=4, random_state=None |

After finalizing the parameters, the machine learning algorithms accurately predicted the results, enhancing their accuracy score. Table 3 provides a comprehensive list of the hyperparameters investigated in our research (*Elgeldawi et al., 2021*). The analysis findings, which also reveal the parameters utilized to generate the excellent matrix score, demonstrate that hyperparameter tuning significantly improved the accuracy of our study's machine-learning models.

# RESULTS AND DISCUSSION

This section examines the exploratory approach and the outcomes of the studies to ascertain the likelihood of survival for heart patients. The results incorporating complete attributes are presented, with a binary classification task utilizing the Death Event attribute to discern whether a patient survived or passed away before the 130-day follow-up period. The SMOTE tool is employed to balance the dataset, and hyperparameter tuning is utilized to improve the forecasting scores of machine learning classifiers. The balanced dataset is then employed to train machine learning algorithms, with evaluations conducted for accuracy, precision, recall, and F1-Score.

## Experiment design

The performance of the algorithms has been examined using supervised ML models. The Python programming language and the Scikit-Learn library module are utilized to create the ML classifiers. The data is divided between the training and testing phases in an 80:20 ratio. Various performance evaluation measures are employed to assess the significance of the ML algorithms. The experiments were conducted entirely in Python, utilizing various library modules from Scikit-Learn. F1 scores, recall, accuracy, and precision were measured using a system with 8 GB of RAM and an Intel(R) Core(TM) m3-7Y30 processor running at 1.00 and 1.61 GHz.

### Performance measuring parameters

The important performance indicators include true positive (TP), true negative (TN), false positive (FP), false negative (FN), F1-score, precision, recall, and accuracy.

- ''TP'' case, in which the value reflects a positive trend for both actual and forecasted values.
- The ''TN'' instance is where the real value is yes, and the forecast value is no.
- When the projected value is yes, and the actual value is no, the situation is directed to an ''FP'' case.
- Forecast value being no and real value being yes in this situation is known as the ''FN'' case.

#### *Accuracy*

The accuracy score of the model that significantly outperforms others highlights its proficiency in clinical forecasting. In a clinical context, the algorithm's defect rate is directly tied to its accuracy, with improved correctness as the fault rate decreases. The degree of accuracy is determined by dividing the number of precise predictions by the total predictions, underscoring its relevance and reliability in clinical applications. The formula for calculating the accuracy score is as follows:

$$Accuracy = \frac{TP + TN}{TP + TN + FP + FN}. \tag{4}$$

#### *Precision*

The precision in this context refers to the accuracy of the methods employed in determining the sample size. This high precision underscores the reliability and accuracy of our approach, particularly in clinical contexts where precise sample size calculations are crucial for ensuring the validity and statistical power of studies. The formula to determine the accuracy precision is as follows:

$$Precision = \frac{TP}{TP + FP}. \tag{5}$$

#### *Recall*

In the context of clinical applications, recall plays a crucial role as it represents the percentage of accurately identified positive cases relative to the total number of characterized instances. A higher recall score is particularly significant in this setting, as it indicates fewer instances of false negatives. This means that the model correctly identifies a greater percentage of positive outcomes, which is crucial for ensuring that potential clinical conditions are not overlooked. The formula for calculating the recall score is as follows:

$$Recall = \frac{TP}{TP + FN}. \tag{6}$$

**Table 4** Compares the measured results of all applied algorithms without using the proposed method.

| Sr.no | Models | Train time (seconds) | Accuracy % | Precision% | Recall % | F1 Score % |
|---|---|---|---|---|---|---|
| 1 | LoR | 0.033 | 87 | 86 | 87 | 86 |
| 2 | RF | 0.142 | 93 | 94 | 93 | 93 |
| 3 | SVM | 0.009 | 72 | 51 | 72 | 60 |
| 4 | DT | 0.005 | 87 | 87 | 87 | 87 |
| 5 | XGB | 0.031 | 93 | 94 | 93 | 93 |
| 6 | GNB | 0.006 | 83 | 83 | 83 | 83 |
| 7 | KNN | 0.003 | 58 | 56 | 58 | 57 |
| 8 | ETC | 0.157 | 77 | 82 | 77 | 70 |
| 9 | GB | 0.068 | 95 | 95 | 95 | 95 |

### *F1 score*

The F1 score is instrumental in quantifying the balance between recall and precision. In the assessment of binary classification models, particularly in clinical contexts, the F1 score serves as a vital statistical indicator. It mandates a cohesive relationship between precision and recall, culminating in the supremacy of the F1 score. In clinical applications, the F1 score proves invaluable for its ability to succinctly capture the trade-off between correctly identified instances and accurately predicted positive cases. The formula to determine the F1 score is listed below:

$$F1 - Score = 2 * \frac{recall * precision}{recall + precision}. \tag{7}$$

## Study results discussion without using proposed technique

Table 4 compares and contrasts the algorithms without using our suggested strategy. All learning algorithms achieved acceptable accuracy and involved timed computations. The GB classifier attained a correctness score of 95 percent, with a precision of 95 percent, a recall of 95 percent, and an F-1 score of 95 percent. Based on measurement metrics and investigation, the KNN model yielded the lowest accuracy at 58 percent, with precision scores of 56 percent, recall scores of 58 percent, and F-1 scores of 57 percent. Regarding computation time, models KNN and DT achieved the lowest training computation times of 0.003 and 0.005, respectively. However, the performance metrics of the tested algorithms indicate that most models do not predict heart failure effectively, as they need to achieve better balance. The classification report analysis of each method is detailed in Table 5.

Figure 7 evaluates the accuracy of all applied machine learning (ML) algorithms. This bar chart-based graph displays the accuracy results of all applied algorithms without using the proposed approach. Figure 8 depicts the K-fold evaluation of model overfitting. This bar chart-based graph shows the evaluation of accuracy scores with k-fold, without using SMOTE.

### *Results validation using K-fold cross validation without SMOTE*

The performance assessment of all algorithms, focusing on addressing overfitting issues, is presented in Table 6. The 10 K-fold cross-validation technique was employed to validate

**Table 5** Class-wise summary report for each model by specific target class without the proposed method.

| Death Event | Precision | Recall | F1-Score | Support |
|---|---|---|---|---|
| LoR | | | | |
| 0 | 0.89 | 0.93 | 0.91 | 43 |
| 1 | 0.80 | 0.71 | 0.75 | 17 |
| RF | | | | |
| 0 | 0.91 | 1.0 | 0.96 | 43 |
| 1 | 1.0 | 0.76 | 0.87 | 17 |
| SVM | | | | |
| 0 | 0.72 | 1.0 | 0.83 | 43 |
| 1 | 0.00 | 0.00 | 0.00 | 17 |
| DT | | | | |
| 0 | 0.93 | 0.88 | 0.90 | 43 |
| 1 | 0.74 | 0.82 | 0.78 | 17 |
| XGB | | | | |
| 0 | 0.91 | 1.00 | 0.96 | 43 |
| 1 | 1.00 | 0.76 | 0.87 | 17 |
| GNB | | | | |
| 0 | 0.85 | 0.93 | 0.89 | 43 |
| 1 | 0.77 | 0.59 | 0.67 | 17 |
| KNN | | | | |
| 0 | 0.70 | 0.74 | 0.72 | 43 |
| 1 | 0.21 | 0.18 | 0.19 | 17 |
| ETC | | | | |
| 0 | 0.75 | 1.0 | 0.86 | 43 |
| 1 | 1.0 | 0.18 | 0.30 | 17 |
| GB | | | | |
| 0 | 0.93 | 1.00 | 0.97 | 43 |
| 1 | 1.00 | 0.82 | 0.90 | 17 |

the robustness of our models. The examination results demonstrated a 95 percent accuracy score using the K-fold approach without employing the SMOTE technique. Figure 8 illustrates that certain algorithms fail to achieve balanced accuracy with the K-fold cross-validation method. Visual inspection indicates that SVM, KNN, ETC classifiers yielded low accuracy scores. The K-fold investigation revealed that all algorithms exhibited signs of overfitting and necessitated rebalancing.

## Study results discussion using SMOTE technique

The performance indicators for the algorithms used in our proposed study are displayed in Table 7. Performance indicator results and computation time analyses were computed using our proposed method. The outcomes demonstrate that all ML algorithms employed to forecast heart failure received the highest performance matrix scores. The outcomes of all the used classifiers are shown in Fig. 9, along with the outcome from our top model, the RF classifier, which attained an excellent correctness score of 96.34 percent for all

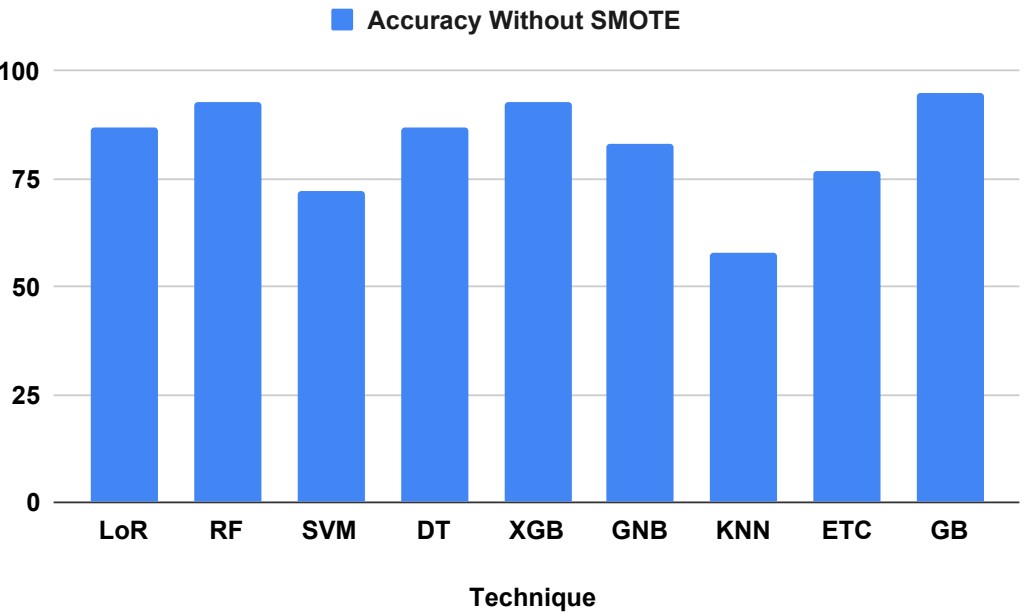

**Figure 7** Shows the accuracy result of all applied algorithms without using the proposed approach.

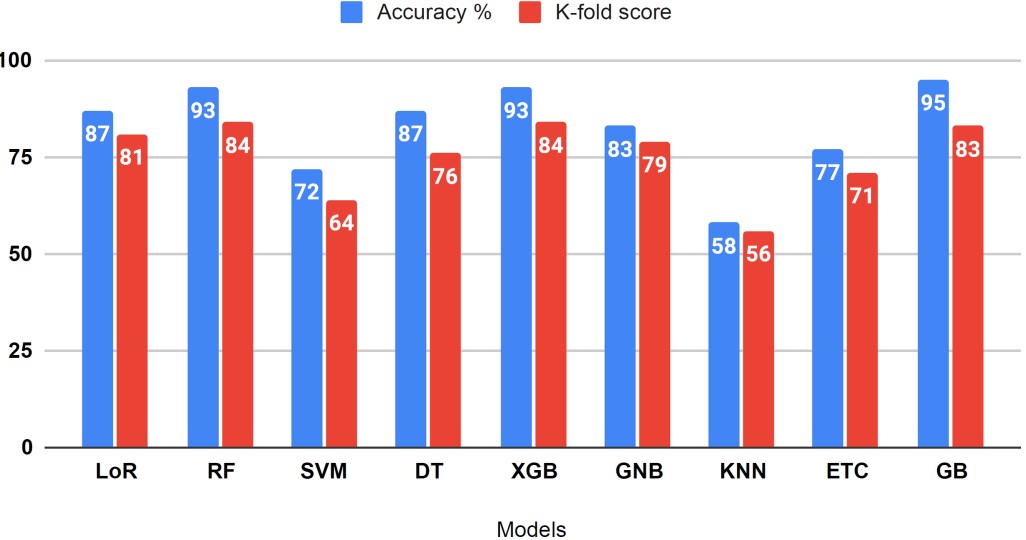

**Figure 8** Showed the evaluation of accuracy score with k-fold without using SMOTE.

the parameters in the confusion matrix. Other models, including the Extreme Gradient Boosting (XGB) and gradient boosting (GB) classifiers, achieved 95 percent, logistic regression (LoR) 82 percent, extra trees classifier (ETC) 91 percent, Gaussian naive Bayes (GNB) 83 percent, and decision tree (DT) 78 percent. All of these models achieved a

**Table 6  Shows the evaluation of the k-fold without using the proposed approach.**

| Sr.no | Models | Accuracy% | 10 K-fold accuracy |
|---|---|---|---|
| 1 | LoR | 87 | 81 |
| 2 | RF | 93 | 84 |
| 3 | SVM | 72 | 64 |
| 4 | DT | 87 | 76 |
| 5 | XGB | 93 | 84 |
| 6 | GNB | 83 | 79 |
| 7 | KNN | 58 | 56 |
| 8 | ETC | 77 | 71 |
| 9 | GB | 95 | 84 |

**Table 7  Using the proposed approach, a competitive analysis method of employed models using SMOTE on test data.**

| Sr.no | Models | Train time (seconds) | Accuracy score % | Precision% | Recall % | F1 score % |
|---|---|---|---|---|---|---|
| 1 | LoR | 0.043 | 82 | 82 | 82 | 82 |
| 2 | RF | 0.452 | 96 | 96 | 96 | 96 |
| 3 | SVM | 0.012 | 57 | 57 | 57 | 57 |
| 4 | DT | 0.005 | 78 | 80 | 78 | 78 |
| 5 | XGB | 0.040 | 95 | 95 | 95 | 95 |
| 6 | GNB | 0.004 | 83 | 83 | 83 | 83 |
| 7 | KNN | 0.003 | 56 | 56 | 56 | 56 |
| 8 | ETC | 0.340 | 91 | 91 | 91 | 91 |
| 9 | GB | 0.202 | 95 | 95 | 95 | 95 |

good accuracy score. The models with the lowest accuracy scores were k-nearest neighbors (KNN) (56 percent) and support vector machine (SVM) (57 percent). We analyzed it and found that the time computation analysis showed the training time for all the models we used in our study. For heart failure prediction, the model that performs the best, RF, gives the highest accuracy score of 96.34 percent in 0.452 s (sec). The LoR train time was 0.043 s, SVM was 0.040 s, DT was 0.005 s, XGB was 0.040 s, ETC was 0.340 s, and GB was 0.202 s. The models with the lowest train times were KNN (0.003 s) and GNB (0.004 s).

### Comparative analysis using K-fold cross validation

Table 8 presents the performance evaluation of all algorithms to address overfitting issues through 10-fold cross-validations. To verify the presence of overfitting in our models, we utilized the 10-fold cross-validation method as illustrated in Table 8. The results of the investigation indicate that the 10-fold approach yielded a 91 percent accuracy score, aligning with our project's methodology.

Figure 10 displays that the relative algorithms exhibit excellent accuracy scores when the K-fold cross-validation method is applied. According to visual analysis, SVM and KNN achieved low accuracy. The k-fold approach is employed to validate all applied algorithms,

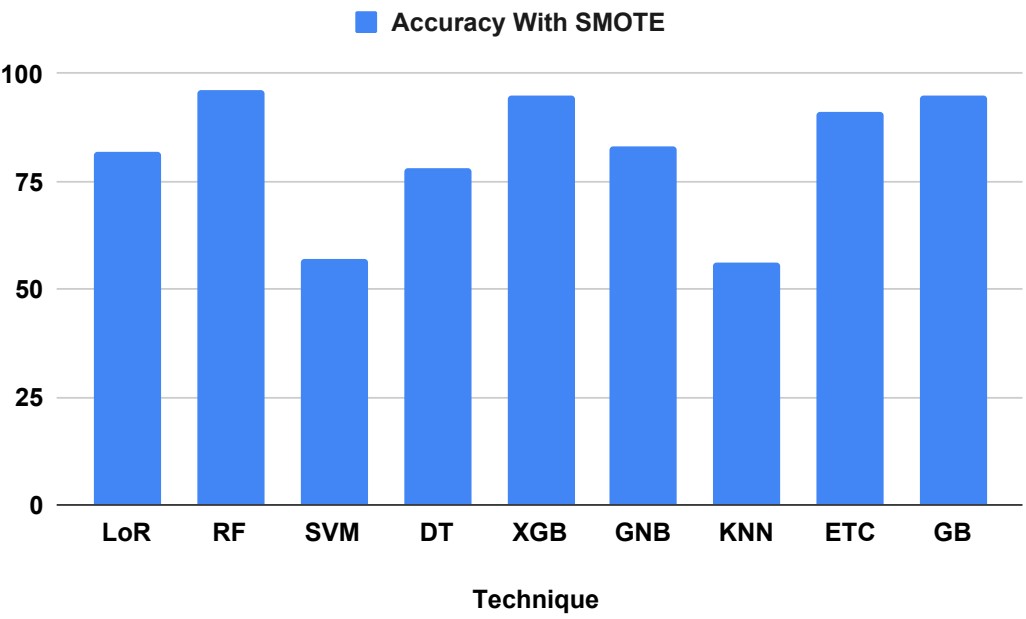

**Figure 9** Showed the accuracy on test data using proposed SMOTE technique.

**Table 8** Analysis of overfitting on applied models using K-fold with our proposed study.

| Sr.no | Models | Accuracy score % | 10-fold accuracy |
|---|---|---|---|
| 1 | LoR | 82 | 77 |
| 2 | RF | 96.34 | 91 |
| 3 | SVM | 57 | 52 |
| 4 | DT | 78 | 83 |
| 5 | XGB | 95 | 90 |
| 6 | GNB | 83 | 81 |
| 7 | KNN | 56 | 55 |
| 8 | ETC | 91 | 88 |
| 9 | GB | 95 | 91 |

and the investigation reveals that all classifiers are balanced, yielding excellent results in test data.

### Classification report results of employed models using SMOTE

For an overview of the target class categorization reports for each model, refer to Table 9. The categorization scores for the models were obtained using the suggested methodology. The investigation's findings demonstrate that GNB and SVM have low accuracy scores in tests of the class metric. Our projected RF model provided an excellent correctness score of 96, while the XGBoost and Gradient Boost classifiers achieved a 95 accuracy score in evaluating classification results. The suggested model for classification outcome evaluation, the Random Forest classifier, attained a 96.34 accuracy score based on the analysis.

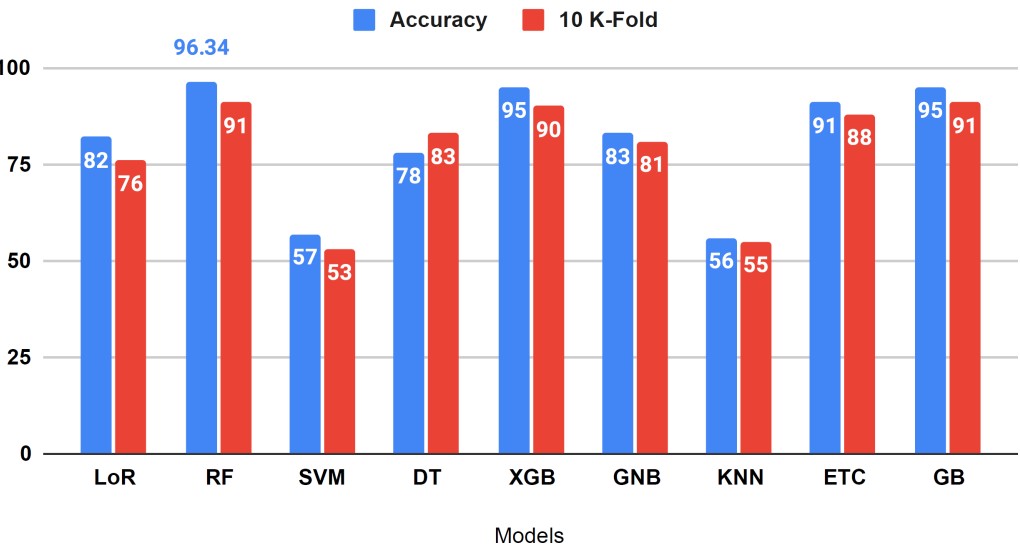

**Accuracy Score and 10 K-Fold score with SMOTE**

**Figure 10** The accuracy score analysis and K-fold method were used to affirm the applied model overfitting.

### Comparison analysis of current study with and without using SMOTE

Figure 11 compares the study results using SMOTE and without SMOTE. In this analysis, the performance of LoR, SVM, and KNN models decreased with the application of SMOTE. However, SMOTE demonstrated effective performance with tree-based algorithms for forecasting HF survival in patients. Figure 11 illustrates the comparative analysis conducted in our study with and without the SMOTE technique.

## Performance analysis using proposed transfer learning features

For an overview of the target class categorization reports for each model, refer to Table 10. The categorization scores for the models were obtained using the suggested methodology. The parameters employed to evaluate our proposed research study include accuracy, precision, recall, and F-1 score. The performance results demonstrate that our proposed study achieved an outstanding correctness score of 97.5 in evaluating classification results. The investigation further substantiates that our proposed transfer learning approach is satisfactory for predicting heart failure survival.

### K-fold cross-validation analysis of proposed transfer learning

The 10-fold cross-validation technique is employed to evaluate the performance of our proposed RF transfer learning model. Table 11 presents the results of the applied 10-fold cross-validation. The findings reveal that the proposed RF model achieved a cross-validation accuracy of 98.7%. Our proposed RF model with transfer learning yields the best outcomes, accompanied by a standard deviation of 0.0164. The analysis indicates that our proposed approach with RF offers a robust analysis result for predicting heart failure survival.

**Table 9  The classification result evaluation of employed models by our SMOTE project method.**

| Death_Event | precision | Recall | F1-score | Support |
|---|---|---|---|---|
| LoR | | | | |
| 0 | 0.80 | 0.82 | 0.81 | 39 |
| 1 | 0.83 | 0.81 | 0.82 | 43 |
| RF | | | | |
| 0 | 0.97 | 0.95 | 0.96 | 39 |
| 1 | 0.95 | 0.98 | 0.97 | 43 |
| SVM | | | | |
| 0 | 0.56 | 0.49 | 0.52 | 39 |
| 1 | 0.58 | 0.65 | 0.62 | 43 |
| DT | | | | |
| 0 | 0.86 | 0.64 | 0.74 | 39 |
| 1 | 0.74 | 0.91 | 0.81 | 43 |
| XGB | | | | |
| 0 | 0.95 | 0.95 | 0.95 | 39 |
| 1 | 0.95 | 0.95 | 0.95 | 43 |
| GNB | | | | |
| 0 | 0.80 | 0.85 | 0.83 | 39 |
| 1 | 0.85 | 0.81 | 0.83 | 43 |
| KNN | | | | |
| 0 | 0.53 | 0.59 | 0.56 | 39 |
| 1 | 0.59 | 0.53 | 0.56 | 43 |
| ETC | | | | |
| 0 | 0.92 | 0.90 | 0.91 | 39 |
| 1 | 0.91 | 0.93 | 0.92 | 43 |
| GB | | | | |
| 0 | 0.97 | 0.92 | 0.95 | 39 |
| 1 | 0.93 | 0.98 | 0.95 | 43 |

**Table 10  The class-wise performance analysis of proposed RF model using transfer learning features.**

| Technique | Accuracy | Target class | Precision | Recall | F1 |
|---|---|---|---|---|---|
| | | 0 | 0.98 | 0.98 | 0.98 |
| RF | 97.5 | 1 | 0.97 | 0.97 | 0.97 |
| | | Average | 0.98 | 0.98 | 0.98 |

### Time complexity analysis of proposed study

The time complexity analysis of the machine learning model RF with the proposed transfer learning technique is presented in Table 12. The analysis indicates that RF requires 0.005 s (sec) for training the transfer features. Notably, RF exhibits outstanding time complexity, completing the model training process in just 0.005 s.

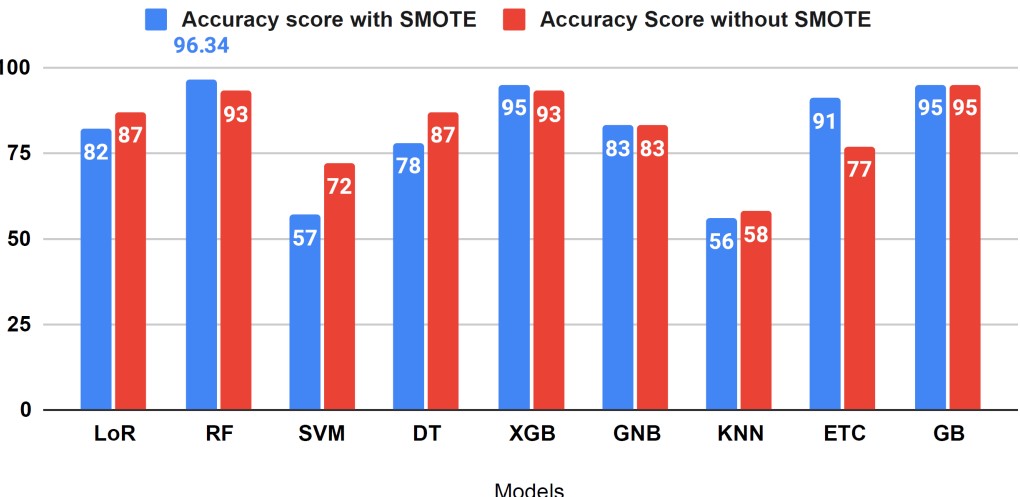

**Accuracy score with SMOTE and Accuracy Score without SMOTE**

**Figure 11** Accuracy score analysis of models with and without SMOTE.

**Table 11** K-fold cross-validation accuracy of the proposed approach to validate model performance.

| Technique | K-fold | K-fold accuracy | Standard deviation (+/-) |
|---|---|---|---|
| Transfer learning (RF) | 10 | 98.7% | 0.0164 |

**Table 12** Runt time computation analysis of proposed approach.

| Technique | Runtime computations (seconds) |
|---|---|
| RF transfer learning | 0.005 |

### Comparison with state of the art studies

We conducted a comparative analysis by juxtaposing our dataset with findings from previous studies, as outlined in Table 13. We have included the studies published in the last two years for comparison. The evaluation criteria encompassed the year of study, approach type, predicted approach, accuracy, precision, recall, and F1 score. Our investigation demonstrated that the random forest model outperformed the earlier study. Notably, our novel approach, the RF+Transfer Learning technique, yielded the most precise results.

### Confusion matrix analysis of our proposed approach

A confusion matrix investigation demonstrates that the results of our performance matrix are accurate, as depicted in Fig. 12. Our proposed approach classifier, RF, which performed effectively, utilizes this matrix. According to the analysis, 42 true negatives and 38 true positives were identified. Only one false negative and one false positive result were observed

**Table 13  Comparison analysis of previously conducted study with the proposed approach.**

| Ref. | Year | Type of approach | Projected technique | Accuracy | Precision | Recall | F1 score |
|------|------|------------------|---------------------|----------|-----------|--------|----------|
| *Ishaq et al. (2021)* | 2021 | Machine learning | Extra tree classifier (ET) | 92.6 | 93 | 93 | 93 |
| **Proposed** | 2023 | Machine learning | Transfer learning (RF) | 97.5 | 98 | 98 | 98 |

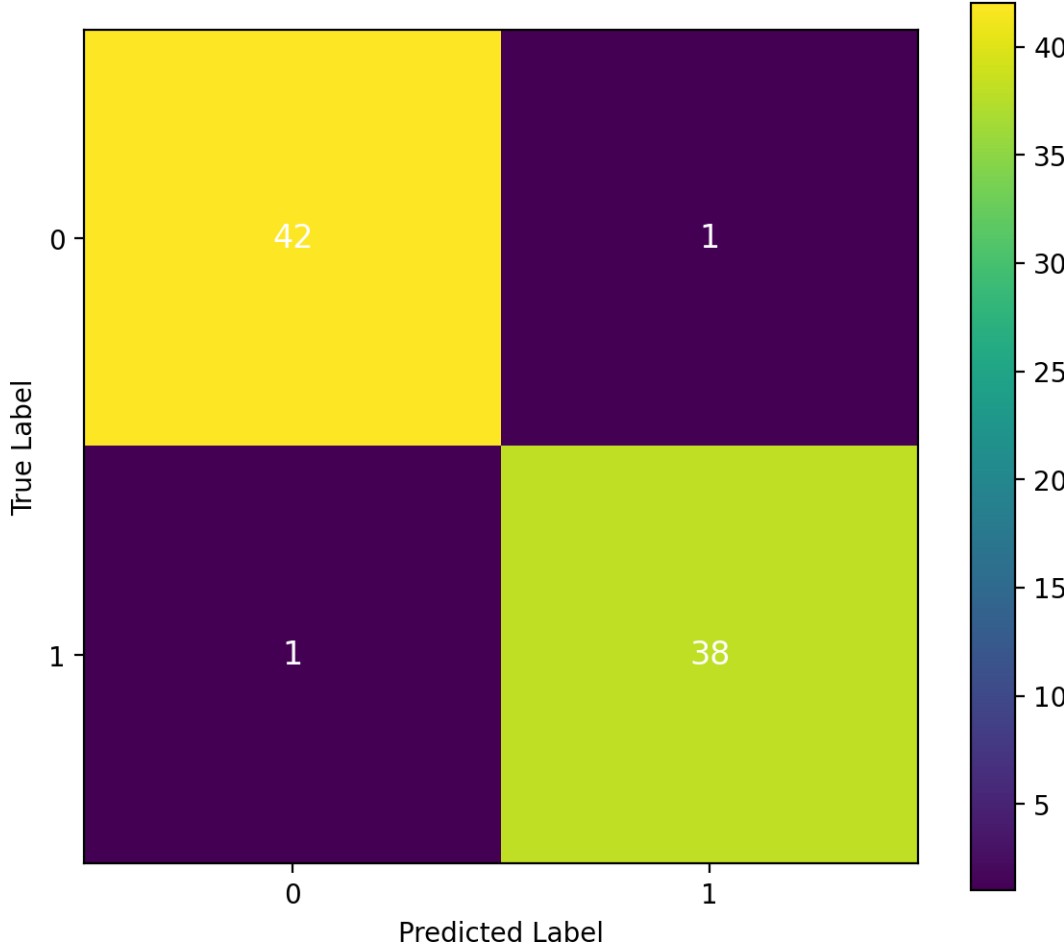

**Figure 12  Confusion matrix analysis and assurance of the proposed approach transfer learning.**

in this study. The confusion matrix has validated our high-performing classifier's accuracy of 97.5 percent in predicting heart failure.

## Discussions and limitations

This research proposed a novel transfer learning-based feature engineering approach, generating a new probabilistic feature set from patient data using the ensemble trees method, Random Forest. The proposed approach achieved a performance accuracy of 97%. However, there is still a 3% error rate. We aim to further improve performance scores by optimizing the architecture of the proposed approach and implementing advanced mechanisms.

# CONCLUSION

Processing raw medical data about the hearts of heart patients using machine learning classifiers has the potential to save lives. Identifying factors that increase the likelihood of heart failure enables the implementation of preventive measures to reduce death rates. This research focuses on forecasting HF survival through ML and utilizes data from 299 clinical record patients. The study employs a novel RF+ transfer learning approach, incorporating nine ML algorithms, namely LoR, RF, SVM, DT, XGB, GNB, KNN, ETC, and GB. To address the class imbalance, the SMOTE tool is applied. Using SMOTE improves the accuracy of tree-like algorithms in predicting the survival of HF patients. The performance metrics for the novel RF+ transfer learning with SMOTE accuracy are measured at 0.97 accuracy, 0.98 precision, 0.98 recall, and 0.98 F1 score. The performance of all employed ML algorithms is analyzed based on the complete set of attributes in the HF dataset. The results indicate that tree-like structure approaches are highly effective in achieving maximum accuracy. The proposed RF outperforms using the novel transfer learning approach with 97 percent accuracy and a computation time of 0.413. A refined study confirms the higher accuracy of our proposed model. Overfitting of the models is investigated using 10-fold cross-validation.

## Future work

Our research study has the potential to propel the medical field forward by aiding doctors in predicting the likelihood of survival for patients with heart failure. Additionally, it will assist healthcare professionals in identifying critical risk factors for those heart failure patients who survive. To enhance the robustness of our research, we plan to address the current limitations by incorporating additional patient data into the dataset. This expansion will involve managing more parameters and employing advanced techniques such as deep learning and feature engineering to improve the accuracy of predicting heart failure outcomes.

## Funding

This work was supported by Atiq ur Rehman, Artificial Intelligence and Intelligent Systems Research Group, School of Innovation, Design and Engineering, Mälardalen University, Västerås, Sweden. The funders had no role in study design, data collection and analysis, decision to publish, or preparation of the manuscript.

## Grant Disclosures

The following grant information was disclosed by the authors:
Atiq ur Rehman, Artificial Intelligence and Intelligent Systems Research Group, School of Innovation, Design and Engineering, Mälardalen University, Västerås, Sweden.

## Competing Interests

The authors declare there are no competing interests.

## Author Contributions

- Azam Mehmood Qadri conceived and designed the experiments, analyzed the data, prepared figures and/or tables, and approved the final draft.
- Muhammad Shadab Alam Hashmi performed the experiments, authored or reviewed drafts of the article, and approved the final draft.
- Ali Raza conceived and designed the experiments, performed the experiments, authored or reviewed drafts of the article, and approved the final draft.
- Syed Ali Jafar Zaidi analyzed the data, prepared figures and/or tables, authored or reviewed drafts of the article, and approved the final draft.
- Atiq ur Rehman performed the computation work, prepared figures and/or tables, and approved the final draft.

## Data Availability

The data is available at Kaggle and the UCI Machine Learning Repository:

- Heart failure clinical records. (2020). UCI Machine Learning Repository. https://doi.org/10.24432/C5Z89R.

- https://www.kaggle.com/datasets/andrewmvd/heart-failure-clinical-data.

## Supplemental Information

Supplemental information for this article can be found online at http://dx.doi.org/10.7717/peerj-cs.1894#supplemental-information.

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
