# Peer review of "Heart failure survival prediction using novel transfer learning based probabilistic features"

_PeerJ Computer Science, doi:10.7717/peerj-cs.1894_

## Round 0.1 · original submission · Major Revisions

The reviewers have substantial concerns about this manuscript. The authors should provide point-to-point responses to address all the concerns and provide a revised manuscript with the revised parts being marked in different color.

·

Basic reporting

The paper discusses the development of a robust survival prediction model for heart failure patients using advanced machine learning techniques. The study analyzed data from 299 hospitalized heart failure patients, addressing the issue of imbalanced data with the Synthetic Minority Oversampling (SMOTE) method. A novel transfer learning-based feature engineering approach was proposed, creating a new probabilistic feature set from patient data using ensemble trees. Nine fine-tuned machine learning models were built and compared to evaluate performance in patient survival prediction. The novel transfer learning mechanism applied to the random forest model outperformed other models, achieving a remarkable accuracy of 0.975. The models underwent evaluation using 10-fold cross-validation and tuning through hyperparameter optimization. The study's findings have the potential to significantly advance cardiovascular medicine by providing more accurate and personalized prognostic assessments for individuals with heart failure. However, there are some issues before publication.

Experimental design

1. (Line 30-31): The introduction briefly mentions coronary artery disease (CAD) as a primary cause of heart failure but doesn't delve into how this relates to the study's focus on survival prediction using machine learning. Expanding on this connection could provide a clearer context for the research.

2. (Line 321-322): While the use of Python and Scikit-Learn is mentioned for creating ML classifiers, additional details on the specific analytical techniques and algorithms used would enhance the reader's understanding of the methodology.

Validity of the findings

3. (Line 313-315): The section on results discusses the outcomes of the survival prediction models. However, a more in-depth discussion on how these results compare with existing methods, particularly in practical healthcare settings, would be beneficial for understanding the study's impact.

4. The paper would benefit from a dedicated section discussing the limitations of the study. This is crucial for providing a balanced view and for guiding future research directions.

5. Future Work (Not Specified): While the paper presents significant findings, outlining potential avenues for future research based on these results would be valuable. This could include suggestions for expanding the dataset, testing other machine learning models, or exploring real-world applications of the model.

Reviewer 2 ·

Basic reporting

The article was written in English, employing clear, unambiguous, and technically correct language while adhering to professional standards of courtesy and expression.

A sufficient introduction and background were provided to illustrate how the work integrates into the broader field of knowledge.

The structure of the article followed an acceptable format. However, there was a lack of clarity in explaining formulas for readers, particularly regarding the confusion between the recall formula and sensitivity. While I recognize that these formulas are commonly used in machine learning, it is crucial to distinguish their application in clinical contexts. Unfortunately, the article did not share raw data.
The figures in the article were pertinent to the content, possessed sufficient resolution, and were appropriately described and labeled.

The results were clear and directly connected to the terms mentioned. However, it was noted that certain definitions, such as Accuracy Score, Precision, Recall, and F1 Score, appeared tailored specifically to machine learning and lacked clarity regarding their correlation to clinical prediction. Further clarification on the statistical relevance of these formulas in the context of clinical predictions is necessary.

Experimental design

If the patient is in the 3rd or 4th stage, what is the impact on diastolic function? Considering that the time frame for stages 3 or 4 of heart failure is typically short, aligning with common clinical understanding, it raises questions about the significance of focusing on these stages in the article. Why not include stages 1 or 2, or explore combinations of different medical treatments? Additionally, it would be valuable to know which medications physicians administered after diagnosing heart failure and the potential clinical implications.
The clarity of the model codes is a concern. While I understand that the authors ran all other nine tuning data before implementing the new model, the article may be statistically significant, but its clinical relevance remains unclear. Providing more details on the methods, especially those that would allow replication, would enhance the article's overall value.
The choice to exclusively use CK as a parameter raises questions. Why not include other markers like troponin, BNP, and ANP? What about factors such as regurgitation rate, valve function, and electrolytes like Mg+ and Ca+? The article's focus on diabetes rather than high cholesterol also warrants clarification.
Exploring aspects beyond the medical realm, such as metabolic equivalents, exercise tolerance tests, lifestyle changes, and quality of life, could provide a more comprehensive understanding of the patient's condition.
In summary, there is a need for greater clarity on the clinical significance of focusing on stages 3 and 4, detailed information about medications and methodology, and a broader consideration of relevant factors beyond CK and diabetes.

Validity of the findings

The article primarily approaches the subject from a computer science standpoint, and while the data were run before implementing the new model, the clinical significance appears to be limited. The efficiency and accuracy, as evident from the tables and figures, seem suboptimal. It would be beneficial to elaborate on the reasons behind the observed low resource efficiency and accuracy.
The novelty of the new model is not clearly articulated for readers, making it challenging to discern its distinct contributions. Additionally, the tables lack clarity, especially when dealing with large age distributions. Understanding the correlation's relevance for prediction becomes crucial when, for instance, what if only one patient is 50 years old, and 80% of patients are 80 years old.
In Figure 2, line 198 states, "Notably, only the feature 'time' displays a strong negative correlation in the dataset." However, there is a similar value for diabetes. The article should provide an explanation for this difference. Please see below:

Table 7: While coincidences can occur, it is notable that all results appear to be the same except for precision. Sharing raw data would help in comprehending the underlying patterns and potential variations.

Additional comments

The conclusions are appropriately stated and tied to the original question investigated. However, it's crucial to acknowledge that the causative relationship is not supported by a well-controlled experimental intervention. Providing additional context or insights into the limitations of the study would strengthen the article.

Annotated reviews are not available for download in order to protect the identity of reviewers who chose to remain anonymous.

Reviewer 3 ·

Basic reporting

Line 98: Replace 'Chi 2 square' to 'Chi-squared'
Line 200: Increase the resolution of figure (a)
Line 290-297: Modify the proposed parameter 'X1', 'Xnew', 'T1', 'Ti' to what you did on Line 297.
Equation 4, 5, 6 & 7: Modify it using latex code: \frac{}{}
Line 383: There is no Figure 15.
Table 11: Replace 'Kfold' to 'K-fold'

Experimental design

No Comment

Validity of the findings

No comment

---

## Round 0.2 · accepted · Accept

Reviewers are satisfied with the revisions, and I concur to recommend accepting this manuscript.

·

Basic reporting

After a thorough review and consideration of all the content, and noting that the author has satisfactorily addressed all comments, the paper is now ready for publication.

Experimental design

After a thorough review and consideration of all the content, and noting that the author has satisfactorily addressed all comments, the paper is now ready for publication.

Validity of the findings

After a thorough review and consideration of all the content, and noting that the author has satisfactorily addressed all comments, the paper is now ready for publication.

Reviewer 2 ·

Basic reporting

The article was written in English and used clear, unambiguous, technically correct text. The article
conform to professional standards of courtesy and expression.

Experimental design

In the 3rd or 4th stage, the study may not have a strong significant value.
The authors have revised the article based on my suggestions except for some data that could be collected to increase the significance.

Validity of the findings

The article primarily approaches the subject from a computer science standpoint, and while the data were
run before implementing the new model, the clinical significance appears to be limited. The efficiency and
accuracy have increased after checking again.